# Planning with Goal-Conditioned Policies

**Soroush Nasiriany**,* **Vitchyr H. Pong**,* **Steven Lin, Sergey Levine**
University of California, Berkeley
{snasiriany,vitchyr,stevenlin598,svlevine@berkeley.edu}

## Abstract

Planning methods can solve temporally extended sequential decision making problems by composing simple behaviors. However, planning requires suitable abstractions for the states and transitions, which typically need to be designed by hand. In contrast, model-free reinforcement learning (RL) can acquire behaviors from low-level inputs directly, but often struggles with temporally extended tasks. Can we utilize reinforcement learning to automatically form the abstractions needed for planning, thus obtaining the best of both approaches? We show that goal-conditioned policies learned with RL can be incorporated into planning, so that a planner can focus on which states to reach, rather than how those states are reached. However, with complex state observations such as images, not all inputs represent valid states. We therefore also propose using a latent variable model to compactly represent the set of valid states for the planner, so that the policies provide an abstraction of actions, and the latent variable model provides an abstraction of states. We compare our method with planning-based and model-free methods and find that our method significantly outperforms prior work when evaluated on image-based robot navigation and manipulation tasks that require non-greedy, multi-staged behavior.

## 1 Introduction

Reinforcement learning can acquire complex skills by learning through direct interaction with the environment, sidestepping the need for accurate modeling and manual engineering. However, complex and temporally extended sequential decision making requires more than just well-honed reactions. Agents that generalize effectively to new situations and new tasks must reason about the consequences of their actions and solve new problems via planning. Accomplishing this entirely with model-free RL often proves challenging, as purely model-free learning does not inherently provide for temporal compositionality of skills. Planning and trajectory optimization algorithms encode this temporal compositionality by design, but require accurate models with which to plan. When these models are specified manually, planning can be very powerful, but learning such models presents major obstacles: in complex environments with high-dimensional observations such as images, direct prediction of future observations presents a very difficult modeling problem [4, 43, 36, 6, 27, 3, 31], and model errors accumulate over time [39], making their predictions inaccurate in precisely those long-horizon settings where we most need the compositionality of planning methods. Can we obtain the benefits temporal compositionality inherent in model-based planning, without the need to model the environment at the lowest level, in terms of both time and state representation?

One way to avoid modeling the environment in detail is to plan over *abstractions*: simplified representations of states and transitions on which it is easier to construct predictions and plans. *Temporal* abstractions allow planning at a coarser time scale, skipping over the high-frequency details and instead planning over higher-level subgoals, while *state* abstractions allow planning over a

---

simpler representation of the state. Both make modeling and planning easier. In this paper, we study how model-free RL can be used to provide such abstraction for a model-based planner. At first glance, this might seem like a strange proposition, since model-free RL methods learn value functions and policies, not models. However, this is precisely what makes them ideal for abstracting away the complexity in temporally extended tasks with high-dimensional observations: by avoiding low-level (e.g., pixel-level) prediction, model-free RL can acquire behaviors that manipulate these low-level observations without needing to predict them explicitly. This leaves the planner free to operate at a higher level of abstraction, reasoning about the capabilities of low-level model-free policies.

Building on this idea, we propose a *model-free* planning framework. For *temporal* abstraction, we learn low-level goal-conditioned policies, and use their value functions as implicit models, such that the planner plans over the goals to pass to these policies. Goal-conditioned policies are policies that are trained to reach a goal state that is provided as an additional input [24, 55, 53, 48]. While in principle such policies can solve any goal-reaching problem, in practice their effectiveness is constrained to nearby goals: for long-distance goals that require planning, they tend to be substantially less effective, as we illustrate in our experiments. However, when these policies are trained together with a value function, as in an actor-critic algorithms, the value function can provide an indication of whether a particular goal is reachable or not. The planner can then plan over intermediate subgoals, using the goal-conditioned value function to evaluate reachability. A major challenge with this setup is the need to actually optimize over these subgoals. In domains with high-dimensional observations such as images, this may require explicitly optimizing over image pixels. This optimization is challenging, as realistic images – and, in general, feasible states – typically form a thin, low-dimensional manifold within the larger space of possible state observation values [34]. To address this, we also build abstractions of the state observation by learning a compact latent variable state representation, which makes it feasible to optimize over the goals in domains with high-dimensional observations, such as images, without explicitly optimizing over image pixels. The learned representation allows the planner to determine which subgoals actually represent feasible states, while the learned goal-conditioned value function tells the planner whether these states are reachable.

Our contribution is a method for combining model-free RL for short-horizon goal-reaching with model-based planning over a latent variable representation of subgoals. We evaluate our method on temporally extended tasks that require multistage reasoning and handling image observations. The low-level goal-reaching policies themselves cannot solve these tasks effectively, as they do not plan over subgoals and therefore do not benefit from temporal compositionality. Planning without state representation learning also fails to perform these tasks, as optimizing directly over images results in invalid subgoals. By contrast, our method, which we call Latent Embeddings for Abstracted Planning (LEAP), is able to successfully determine suitable subgoals by searching in the latent representation space, and then reach these subgoals via the model-free policy.

## 2 Related Work

Goal-conditioned reinforcement learning has been studied in a number of prior works [24, 25, 37, 18, 53, 2, 48, 57, 40, 59]. While goal-conditioned methods excel at training policies to greedily reach goals, they often fail to solve long-horizon problems. Rather than proposing a new goal-conditioned RL method, we propose to use goal-conditioned policies as the abstraction for planning in order to handle tasks with a longer horizon.

Model-based planning in deep reinforcement learning is a well-studied problem in the context of low-dimensional state spaces [50, 32, 39, 7]. When the observations are high-dimensional, such as images, model errors for direct prediction compound quickly, making model-based RL difficult [15, 13, 5, 14, 26]. Rather than planning directly over image observations, we propose to plan at a temporally-abstract level by utilizing goal-conditioned policies. A number of papers have studied embedding high-dimensional observations into a low-dimensional latent space for planning [60, 16, 62, 22, 29]. While our method also plans in a latent space, we additionally use a model-free goal-conditioned policy as the abstraction to plan over, allowing our method to plan over temporal abstractions rather than only state abstractions.

Automatically setting subgoals for a low-level goal-reaching policy bears a resemblance to hierarchical RL, where prior methods have used model-free learning on top of goal-conditioned policies [10, 61,

12, 58, 33, 20, 38]. By instead using a planner at the higher level, our method can flexibly plan to solve new tasks and benefit from the compositional structure of planning.

Our method builds on temporal difference models [48] (TDMs), which are finite-horizon, goal-conditioned value functions. In prior work, TDMs were used together with a single-step planner that optimized over a single goal, represented as a low-dimensional ground truth state (under the assumption that all states are valid) [48]. We also use TDMs as implicit models, but in contrast to prior work, we plan over multiple subgoals and demonstrate that our method can perform temporally extended tasks. More critically, our method also learns abstractions of the state, which makes this planning process much more practical, as it does not require assuming that all state vectors represent feasible states. Planning with goal-conditioned value functions has also been studied when there are a discrete number of predetermined goals [30] or skills [1], in which case graph-search algorithms can be used to plan. In this paper, we not only provide a concrete instantiation of planning with goal-conditioned value functions, but we also present a new method for scaling this planning approach to images, which reside in a lower-dimensional manifold.

Lastly, we note that while a number of papers have studied how to combine model-free and model-based methods [54, 41, 23, 56, 44, 51, 39], our method is substantially different from these approaches: we study how to use model-free policies *as the abstraction for planning*, rather than using models [54, 41, 23, 39] or planning-inspired architectures [56, 44, 51, 21] to accelerate model-free learning.

## 3 Background

We consider a finite-horizon, goal-conditioned Markov decision process (MDP) defined by a tuple $(\mathcal{S}, \mathcal{G}, \mathcal{A}, p, R, T_{\max}, \rho_0, \rho_g)$, where $\mathcal{S}$ is the set of states, $\mathcal{G}$ is the set of goals, $\mathcal{A}$ is the set of actions, $p(\mathbf{s}_{t+1} \mid \mathbf{s}_t, \mathbf{a}_t)$ is the time-invariant (unknown) dynamics function, $R$ is the reward function, $T_{\max}$ is the maximum horizon, $\rho_0$ is the initial state distribution, and $\rho_g$ is the goal distribution. The objective in goal-conditioned RL is to obtain a policy $\pi(\mathbf{a}_t \mid \mathbf{s}_t, \mathbf{g}, t)$ to maximize the expected sum of rewards $\mathbb{E}[\sum_{t=0}^{T_{\max}} R(\mathbf{s}_t, \mathbf{g}, t)]$, where the goal is sampled from $\rho_g$ and the states are sampled according to $\mathbf{s}_0 \sim \rho_0$, $\mathbf{a}_t \sim \pi(\mathbf{a}_t \mid \mathbf{s}_t, \mathbf{g}, t)$, and $\mathbf{s}_{t+1} \sim p(\mathbf{s}_{t+1} \mid \mathbf{s}_t, \mathbf{a}_t)$. We consider the case where goals reside in the same space as states, i.e., $\mathcal{G} = \mathcal{S}$.

An important quantity in goal-conditioned MDPs is the goal-conditioned value function $V^\pi$, which predicts the expected sum of future rewards, given the current state $\mathbf{s}$, goal $\mathbf{g}$, and time $t$:

$$V^\pi(\mathbf{s}, \mathbf{g}, t) = \mathbb{E}\left[\sum_{t'=t}^{T_{\max}} R(\mathbf{s}_{t'}, \mathbf{g}, t') \mid \mathbf{s}_t = \mathbf{s}, \pi \text{ is conditioned on } \mathbf{g}\right].$$

To keep the notation uncluttered, we will omit the dependence of $V$ on $\pi$. While various time-varying reward functions can be used, temporal difference models (TDMs) [48] use the following form:

$$R_{\text{TDM}}(\mathbf{s}, \mathbf{g}, t) = -\delta(t = T_{\max})d(\mathbf{s}, \mathbf{g}). \tag{1}$$

where $\delta$ is the indicator function, and the distance function $d$ is defined by the task. This particular choice of reward function gives a TDM the following interpretation: given a state $\mathbf{s}$, how close will the goal-conditioned policy $\pi$ get to $\mathbf{g}$ after $t$ time steps of attempting to reach $\mathbf{g}$? TDMs can thus be used as a measure of reachability by quantifying how close to another state the policy can get in $t$ time steps, thus providing *temporal* abstraction. However, TDMs will only produce reasonable reachability predictions for *valid* goals – goals that resemble the kinds of states on which the TDM was trained. This important limitation requires us to also utilize *state* abstractions, limiting our search to valid states. In the next section, we will discuss how we can use TDMs in a planning framework over high-dimensional state observations such as images.

## 4 Planning with Goal-Conditioned Policies

We aim to learn a model that can solve arbitrary long-horizon goal reaching tasks with high-dimensional observation and goal spaces, such as images. A model-free goal-conditioned reinforcement learning algorithm could, in principle, solve such a problem. However, as we will show in our experiments, in practice such methods produce overly greedy policies, which can accomplish short-term goals, but struggle with goals that are more temporally extended. We instead combine

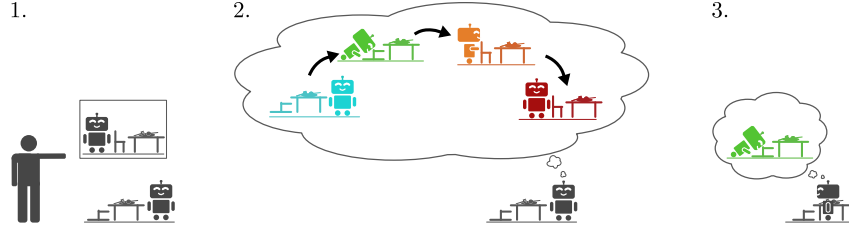

Figure 1: Summary of Latent Embeddings for Abstracted Planning (LEAP). (1) The planner is given a goal state. (2) The planner plans intermediate subgoals in a low-dimensional latent space. By planning in this latent space, the subgoals correspond to valid state observations. (3) The goal-conditioned policy then tries to reach the first subgoal. After $t_1$ time steps, the policy replans and repeats steps 2 and 3.

goal-conditioned policies trained to achieve subgoals with a planner that decomposes long-horizon goal-reaching tasks into $K$ shorter horizon subgoals. Specifically, our planner chooses the $K$ sub-goals, $\mathbf{g}_1, \ldots, \mathbf{g}_K$, and a goal-reaching policy then attempts to reach the first subgoal $\mathbf{g}_1$ in the first $t_1$ time steps, before moving onto the second goal $\mathbf{g}_2$, and so forth, as shown in Figure 1. This procedure only requires training a goal-conditioned policy to solve short-horizon tasks. Moreover, by planning appropriate subgoals, the agent can compose previously learned goal-reaching behavior to solve new, temporally extended tasks. The success of this approach will depend heavily on the choice of subgoals. In the sections below, we outline how one can measure the quality of the subgoals. Then, we address issues that arise when optimizing over these subgoals in high-dimensional state spaces such as images. Lastly, we summarize the overall method and provide details on our implementation.

## 4.1 Planning over Subgoals

Suitable subgoals are ones that are reachable: if the planner can choose subgoals such that each subsequent subgoal is reachable given the previous subgoal, then it can reach any goal by ensuring the last subgoal is the true goal. If we use a goal-conditioned policy to reach these goals, how can we quantify how reachable these subgoals are?

One natural choice is to use a goal-conditioned value function which, as previously discussed, provides a measure of reachability. In particular, given the current state $\mathbf{s}$, a policy will reach a goal $\mathbf{g}$ after $t$ time steps if and only if $V(\mathbf{s}, \mathbf{g}, t) = 0$. More generally, given $K$ intermediate subgoals $\mathbf{g}_{1:K} = \mathbf{g}_1, \ldots, \mathbf{g}_K$ and $K + 1$ time intervals $t_1, \ldots, t_{K+1}$ that sum to $T_{\max}$, we define the *feasibility vector* as

$$\overrightarrow{\mathbf{V}}(\mathbf{s}, \mathbf{g}_{1:K}, t_{1:K+1}, \mathbf{g}) = \begin{bmatrix} V(\mathbf{s}, \mathbf{g}_1, t_1) \\ V(\mathbf{g}_1, \mathbf{g}_2, t_2) \\ \vdots \\ V(\mathbf{g}_{K-1}, \mathbf{g}_K, t_K) \\ V(\mathbf{g}_K, \mathbf{g}, t_{K+1}) \end{bmatrix}.$$

The feasibility vector provides a quantitative measure of a plan's feasibility: The first element describes how close the policy will reach the first subgoal, $\mathbf{g}_1$, starting from the initial state, $\mathbf{s}$. The second element describes how close the policy will reach the second subgoal, $\mathbf{g}_2$, starting from the first subgoal, and so on, until the last term measures the reachability to the true goal, $\mathbf{g}$.

To create a feasible plan, we would like each element of this vector to be zero, and so we minimize the norm of the feasibility vector:

$$\mathcal{L}(\mathbf{g}_{1:K}) = ||\overrightarrow{\mathbf{V}}(\mathbf{s}, \mathbf{g}_{1:K}, t_{1:K+1}, \mathbf{g})||. \qquad (2)$$

In other words, minimizing Equation 2 searches for subgoals such that the overall path is feasible and terminates at the true goal. In the next section, we turn to optimizing Equation 2 and address issues that arise in high-dimensional state spaces.

## 4.2 Optimizing over Images

We consider image-based environments, where the set of states $\mathcal{S}$ is the set of valid image observations in our domain. In image-based environments, solving the optimization in Equation 2 presents two

problems. First, the optimization variables $\mathbf{g}_{1:K}$ are very high-dimensional – even with 64x64 images and just 3 subgoals, there are over 10,000 dimensions. Second, and perhaps more subtle, the optimization iterates must be constrained to the set of valid image observations $\mathcal{S}$ for the subgoals to correspond to meaningful states. While a plethora of constrained optimization methods exist, they typically require knowing the set of valid states [42] or being able to project onto that set [46]. In image-based domains, the set of states $\mathcal{S}$ is an unknown $r$-dimensional manifold embedded in a higher-dimensional space $\mathbb{R}^N$, for some $N \gg r$ [34] – i.e., the set of valid image observations.

Optimizing Equation 2 would be much easier if we could directly optimize over the $r$ dimensions of the underlying representation, since $r \ll N$, and crucially, since we would not have to worry about constraining the planner to an unknown manifold. While we may not know the set $\mathcal{S}$ a priori, we can learn a latent-variable model with a compact latent space to capture it, and then optimize in the latent space of this model. To this end, we use a variational-autoencoder (VAE) [28, 52], which we train with images randomly sampled from our environment.

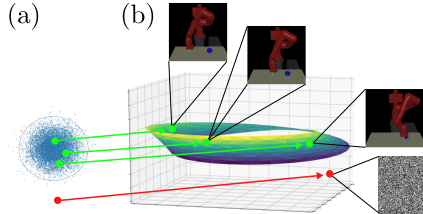

(a)      (b)

A VAE consists of an encoder $q_\phi(\mathbf{z} \mid \mathbf{s})$ and decoder $p_\theta(\mathbf{s} \mid \mathbf{z})$. The inference network maps high-dimensional states $\mathbf{s} \in \mathcal{S}$ to a distribution over lower-dimensional latent variables $\mathbf{z}$ for some lower dimensional space $\mathcal{Z}$, while the generative model reverses this mapping. Moreover, the VAE is trained so that the marginal distribution of $\mathcal{Z}$ matches our prior distribution $p_0$, the standard Gaussian.

Figure 2: Optimizing directly over the image manifold (b) is challenging, as it is generally unknown and resides in a high-dimensional space. We optimize over a latent state (a) and use our decoder to generate images. So long as the latent states have high likelihood under the prior (green), they will correspond to realistic images, while latent states with low likelihood (red) will not.

This last property of VAEs is crucial, as it allows us to tractably optimize over the manifold of valid states $\mathcal{S}$. So long as the latent variables have high likelihood under the prior, the corresponding images will remain inside the manifold of valid states, as shown in Figure 2. In fact, Dai and Wipf [9] showed that a VAE with a Gaussian prior can always recover the true manifold, making this choice for latent-variable model particularly appealing.

In summary, rather than minimizing Equation 2, which requires optimizing over the high-dimensional, unknown space $\mathcal{S}$ we minimize

$$\mathcal{L}_{\text{LEAP}}(\mathbf{z}_{1:K}) = ||\overrightarrow{\mathbf{V}}(\mathbf{s}, \mathbf{z}_{1:K}, t_{1:K+1}, \mathbf{g})||_p - \lambda \sum_{k=1}^{K} \log p(\mathbf{z}_k) \tag{3}$$

where

$$\overrightarrow{\mathbf{V}}(\mathbf{s}, \mathbf{z}_{1:K}, t_{1:K+1}, \mathbf{g}) = \begin{bmatrix} V(\mathbf{s}, \psi(\mathbf{z}_1), t_1) \\ V(\psi(\mathbf{z}_1), \psi(\mathbf{z}_2), t_2) \\ \vdots \\ V(\psi(\mathbf{z}_{K-1}), \psi(\mathbf{z}_K), t_K) \\ V(\psi(\mathbf{z}_K), \mathbf{g}, t_{K+1}) \end{bmatrix} \quad \text{and} \quad \psi(\mathbf{z}) = \arg\max_{\mathbf{g}'} p_\theta(\mathbf{g}' \mid \mathbf{z}).$$

This procedure optimizes over latent variables $\mathbf{z}_k$, which are then mapped onto high-dimensional goal states $\mathbf{g}_k$ using the maximum likelihood estimate (MLE) of the decoder $\arg\max_{\mathbf{g}}(\mathbf{g} \mid \mathbf{z})$. In our case, the MLE can be computed in closed form by taking the mean of the decoder. The term summing over $\log p(\mathbf{z}_k)$ penalizes latent variables that have low likelihood under the prior $p$, and $\lambda$ is a hyperparameter that controls the importance of this second term.

While any norm could be used, we used the $\ell_\infty$-norm which forces each element of the feasibility vector to be near zero. We found that the $\ell_\infty$-norm outperformed the $\ell_1$-norm, which only forces the sum of absolute values of elements near zero. [2]

## 4.3  Goal-Conditioned Reinforcement Learning

For our goal-conditioned reinforcement learning algorithm, we use temporal difference models (TDMs) [48]. TDMs learn Q functions rather that V functions, and so we compute $V$ by evaluating

$Q$ with the action from the deterministic policy: $V(\mathbf{s}, \mathbf{g}, t) = Q(\mathbf{s}, \mathbf{a}, \mathbf{g}, t)|_{\mathbf{a}=\pi(\mathbf{s}, \mathbf{g}, t)}$. To further improve the efficiency of our method, we can also utilize the same VAE that we use to recover the latent space for planning as a state representation for TDMs. While we could train the reinforcement learning agents from scratch, this can be expensive in terms of sample efficiency as much of the learning will focus on simply learning good convolution filters. We therefore use the pretrained mean-encoder of the VAE as the state encoder for our policy and value function networks, and only train additional fully-connected layers with RL on top of these representations. Details of the architecture are provided in Appendix C. We show in Section 5 that our method works without reusing the VAE mean-encoder, and that this parameter reuse primarily helps with increasing the speed of learning.

## 4.4 Summary of Latent Embeddings for Abstracted Planning

Our overall method is called Latent Embeddings for Abstracted Planning (LEAP) and is summarized in Algorithm 1. We first train a goal-conditioned policy and a variational-autoencoder on randomly collected states. Then at testing time, given a new goal, we choose subgoals by minimizing Equation 3. Once the plan is chosen, the first goal $\psi(\mathbf{z}_1)$ is given to the policy. After $t_1$ steps, we repeat this procedure: we produce a plan with $K - 1$ (rather than $K$) subgoals, and give the first goal to the policy. In this work, we fix the time intervals to be evenly spaced out (i.e., $t_1 = t_2 \ldots t_{K+1} = \lfloor T_{\max}/(K+1) \rfloor$), but additionally optimizing over the time intervals would be a promising future extension.

---

**Algorithm 1** Latent Embeddings for Abstracted Planning (LEAP)

---

1: Train VAE encoder $q_\phi$ and decoder $p_\theta$.
2: Train TDM policy $\pi$ and value function $V$.
3: Initialize state, goal, and time: $\mathbf{s}_1 \sim \rho_0$, goal $\mathbf{g} \sim \rho_g$, and $t = 1$.
4: Assign the last subgoal to the true goal, $\mathbf{g}_{K+1} = \mathbf{g}$
5: **for** $k$ in $1, \ldots, K + 1$ **do**
6:     Optimize Equation 3 to choose latent subgoals $\mathbf{z}_k, \ldots, \mathbf{z}_K$ using $V$ and $p_\theta$ if $k \leq K$.
7:     Decode $\mathbf{z}_k$ to obtain goal $\mathbf{g}_k = \psi(\mathbf{z}_k)$.
8:     **for** $t'$ in $1, \ldots, t_k$ **do**
9:         Sample next action $\mathbf{a}_t$ using goal-conditioned policy $\pi(\cdot \mid \mathbf{s}_t, \mathbf{g}_k, t_k - t')$.
10:        Execute $\mathbf{a}_t$ and obtain next state $\mathbf{s}_{t+1}$
11:        Increment the global timer $t \leftarrow t + 1$.
12:     **end for**
13: **end for**

---

## 5 Experiments

Our experiments study the following two questions: **(1)** How does LEAP compare to model-based methods, which directly predict each time step, and model-free RL, which directly optimizes for the final goal? **(2)** How does the use of a latent state representation and other design decisions impact the performance of LEAP?

### 5.1 Vision-based Comparison and Results

We study the first question on two distinct vision-based tasks, each of which requires temporally-extended planning and handling high-dimensional image observations.

The first task, *2D Navigation* requires navigating around a U-shaped wall to reach a goal, as shown in Figure 3. The state observation is a top-down image of the environment. We use this task to conduct ablation studies that test how each component of LEAP contributes to final performance. We also use this environment to generate visualizations that help us better understand how our method uses the goal-conditioned value function to evaluate reachability over images. While visually simple, this task is far from trivial for goal-conditioned and planning methods: a greedy goal-reaching policy that moves directly towards the goal will never reach the goal. The agent must plan a temporally-extended path that moves around the walls, sometimes moving away from the goal. We also use this environment to compare our method with prior work on goal-conditioned and model-based RL.

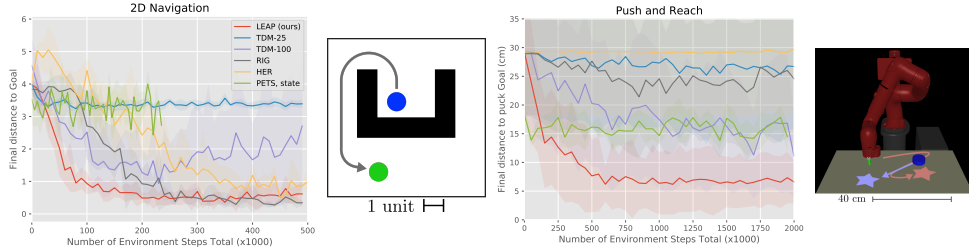

Figure 3: Comparisons on two vision-based domains that evaluate temporally extended control, with illustrations of the tasks. In 2D Navigation (left), the goal is to navigate around a U-shaped wall to reach the goal. In the Push and Reach manipulation task (right), a robot must first push a puck to a target location (blue star), which may require moving the hand away from the goal hand location, and then move the hand to another location (red star). Curves are averaged over multiple seeds and shaded regions represent one standard deviation. Our method, shown in red, outperforms prior methods on both tasks. On the Push and Reach task, prior methods typically get the hand close to the right location, but perform much worse at moving the puck, indicating an overly greedy strategy, while our approach succeeds at both.

To evaluate LEAP on a more complex task, we utilize a robotic manipulation simulation of a *Push and Reach* task. This task requires controlling a simulated Sawyer robot to both (1) move a puck to a target location and (2) move its end effector to a target location. This task is more visually complex, and requires more temporally extended reasoning. The initial arm and and puck locations are randomized so that the agent must decide how to reposition the arm to reach around the object, push the object in the desired direction, and then move the arm to the correct location, as shown in Figure 3. A common failure case for model-free policies in this setting is to adopt an overly greedy strategy, only moving the arm to the goal while ignoring the puck.

We train all methods on randomly initialized goals and initial states. However, for evaluation, we intentionally select difficult start and goal states to evaluate long-horizon reasoning. For 2D Navigation, we initialize the policy randomly inside the center square and sample a goal from the region directly below the U-shaped wall. This requires initially moving away from the goal to navigate around the wall. For Push and Reach, we evaluate on 5 distinct challenging configurations, each requiring the agent to first plan to move the puck, and then move the arm only once the puck is in its desired location. In one configuration for example, we initialize the hand and puck on opposite sides of the workspace and set goals so that the hand and puck must switch sides.

We compare our method to both model-free methods and model-based methods that plan over learned models. All of our tasks use $T_{\max} = 100$, and LEAP uses CEM to optimize over $K = 3$ subgoals, each of which are $25$ time steps apart. We compare directly with model-free TDMs, which we label **TDM-25**. Since the task is evaluated on a horizon of length $T_{\max} = 100$ we also compare to a model-free TDM policy trained for $T_{\max} = 100$, which we label **TDM-100**. We compare to reinforcement learning with imagined goals (**RIG**) [40], a state-of-the-art method for solving image-based goal-conditioned tasks. RIG learns a reward function from images rather than using a pre-determined reward function. We found that providing RIG with the same distance function as our method improves its performance, so we use this stronger variant of RIG to ensure a fair comparison. In addition, we compare to hindsight experiment replay (**HER**) [2] which uses sparse, indicator rewards. Lastly, we compare to probabilistic ensembles with trajectory sampling (PETS) [7], a state-of-the-art model-based RL method. We favorably implemented PETS on the ground-truth low-dimensional state representation and label it **PETS, state**.

The results are shown in Figure 3. LEAP significantly outperforms prior work on both tasks, particularly on the harder Push and Reach task. While the TDM used by LEAP (TDM-25) performs poorly by itself, composing it with 3 different subgoals using LEAP results in much better performance. By 400k environment steps, LEAP already achieves a final puck distance of under 10 cm, while the next best method, TDM-100, requires 5 times as many samples. Details on each task are in Appendix B, and algorithm implementation details are given in Appendix C.

We visualize the subgoals chosen by LEAP in Figure 4 by decoding the latent subgoals $\mathbf{z}_{t_{1:K}}$ into images with the VAE decoder $p_\theta$. In Push and Reach, these images correspond to natural subgoals for the task. Figure 4 also shows a visualization of the value function, which is used by the planner to determine reachability. Note that the value function generally recognizes that the wall is impassable,

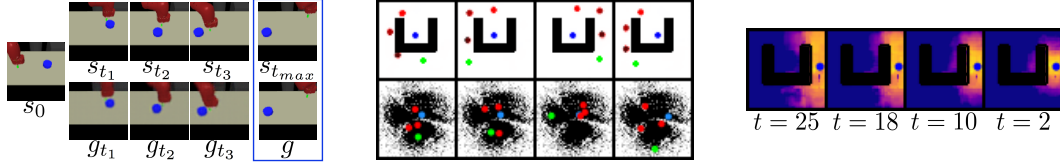

Figure 4: (Left) Visualization of subgoals reconstructed from the VAE (bottom row), and the actual images seen when reaching those subgoals (top row). Given an initial state $s_0$ and a goal image $\mathbf{g}$, the planner chooses meaningful subgoals: at $\mathbf{g}_{t_1}$, it moves towards the puck, at $\mathbf{g}_{t_2}$ it begins pushing the puck, and at $\mathbf{g}_{t_3}$ it completes the pushing motion before moving to the goal hand position at $\mathbf{g}$. (Middle) The top row shows the image subgoals superimposed on one another. The blue circle is the starting position, the green circle is the target position, and the intermediate circles show the progression of subgoals (bright red is $\mathbf{g}_{t_1}$, brown is $\mathbf{g}_{t_3}$). The colored circles show the subgoals in the latent space (bottom row) for the two most active VAE latent dimensions, as well as samples from the VAE aggregate posterior [35]. (Right) Heatmap of the value function $V(\mathbf{s}, \mathbf{g}, t)$, with each column showing a different time horizon $t$ for a fixed state $\mathbf{s}$. Warmer colors show higher value. Each image indicates the value function for all possible goals $g$. As the time horizon decreases, the value function recognizes that it can only reach nearby goals.

and makes reasonable predictions for different time horizons. Videos of the final policies and generated subgoals and code for our implementation of LEAP are available on the paper website[3].

## 5.2 Planning in Non-Vision-based Environments with Unknown State Spaces

While LEAP was presented in the context of optimizing over images, we also study its utility in non-vision based domains. Specifically, we compare LEAP to prior works on an *Ant Navigation* task, shown in Figure 5, where the state-space consists of the quadruped robot's joint angles, joint velocity, and center of mass. While this state space is more compact than images, only certain combinations of state values are actually valid, and the obstacle in the environment is unknown to the agent, meaning that a naïve optimization over the state space can easily result in invalid states (e.g., putting the robot inside an obstacle).

This task has a significantly longer horizon of $T_{\max} = 600$, and LEAP uses CEM to optimize over $K = 11$ subgoals, each of which are $50$ time steps apart. As in the vision-based comparisons, we compare with model-free TDMs, for the short-horizon setting (**TDM-50**) which LEAP is built on top of, and the long-horizon setting (**TDM-600**). In addition to **HER**, we compare to a variant of HER that uses the same rewards and relabeling strategy as RIG, which we label **HER+**. We exclude the PETS baseline, as it has been unable to solve long-horizon tasks such as ours. In this section, we add a comparison to hierarchical reinforcement learning with off-policy correction (**HIRO**) [38], a hierarchical method for state-based goals. We evaluate all baselines on a challenging configuration of the task in which the ant must navigate from one corner of the maze to the other side, by going around a long wall. The desired behavior will result in large negative rewards during the trajectory, but will result in an optimal final state. We see that in Figure 5, LEAP is the only method that successfully navigates the ant to the goal. HIRO, HER, HER+ don't attempt to go around the wall at all, as doing so will result in a large sum of negative rewards. TDM-50 has a short horizon that results in greedy behavior, while TDM-600 fails to learn due to temporal sparsity of the reward.

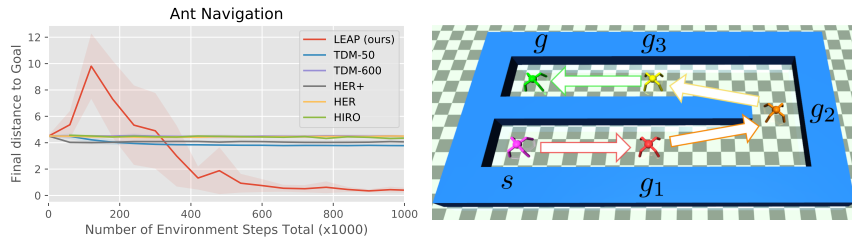

Figure 5: In the Ant Navigation task, the ant must move around the long wall, which will incur large negative rewards during the trajectory, but will result in an optimal final state. We illustrate the task, with the purple ant showing the starting state and the green ant showing the goal. We use 3 subgoals here for illustration. Our method (shown in red in the plot) is the only method that successfully navigates the ant to the goal.

[3]https://sites.google.com/view/goal-planning

### 5.3 Ablation Study

We analyze the importance of planning in the latent space, as opposed to image space, on the navigation task. For comparison, we implement a planner that directly optimizes over image subgoals (i.e., in pixel space). We also study the importance of reusing the pretrained VAE encoder by replicating the experiments with the RL networks trained from scratch. We see in Figure 6 that a model that does not reuse the VAE encoder does succeed, but takes much longer. More importantly, planning over latent states achieves dramatically better performance than planning over raw images. Figure 6 also shows the intermediate subgoals outputted by our optimizer when optimizing over images. While these subgoals may have high value according to Equation 2, they clearly do not correspond to valid state observations, indicating that the planner is exploiting the value function by choosing images far outside the manifold of valid states.

We include further ablations in Appendix A, in which we study the sensitivity of $\lambda$ in Equation 3 (Subsection A.3), the choice of norm (Subsection A.1), and the choice of optimizer (Subsection A.2). The results show that LEAP works well for a wide range of $\lambda$, that $\ell_\infty$-norm performs better, and that CEM consistently outperforms gradient-based optimizers, both in terms of optimizer loss and policy performance.

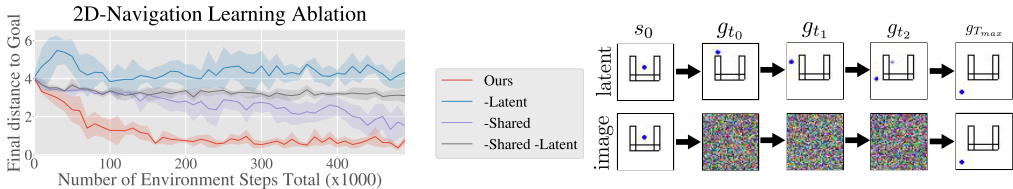

Figure 6: (Left) Ablative studies on 2D Navigation. We keep all components of LEAP the same but replace optimizing over the latent space with optimizing over the image space (-latent). We separately train the RL methods from scratch rather than reusing the VAE mean encoder (-shared), and also test both ablations together (-latent, shared). We see that sharing the encoder weights with the RL policy results in faster learning, and that optimizing over the latent space is critical for success of the method. (Right) Visualization of the subgoals generated when optimizing over the latent space and decoding the image (top) and when optimizing over the images directly (bottom). The goals generated when planning in image space are not meaningful, which explains the poor performance of "-latent" shown in (Left).

## 6   Discussion

We presented Latent Embeddings for Abstracted Planning (LEAP), an approach for solving temporally extended tasks with high-dimensional state observations, such as images. The key idea in LEAP is to form *temporal* abstractions by using goal-reaching policies to evaluate reachability, and *state* abstractions by using representation learning to provide a convenient state representation for planning. By planning over states in a learned latent space and using these planned states as subgoals for goal-conditioned policies, LEAP can solve tasks that are difficult to solve with conventional model-free goal-reaching policies, while avoiding the challenges of modeling low-level observations associated with fully model-based methods. More generally, the combination of model-free RL with planning is an exciting research direction that holds the potential to make RL methods more flexible, capable, and broadly applicable. Our method represents a step in this direction, though many crucial questions remain to be answered. Our work largely neglects the question of exploration for goal-conditioned policies, and though this question has been studied in some recent works [17, 45, 59, 49], examining how exploration interacts with planning is an exciting future direction. Another exciting direction for future work is to study how lossy state abstractions might further improve the performance of the planner, by explicitly discarding state information that is irrelevant for higher-level planning.

## 7   Acknowledgments

This work was supported by the Office of Naval Research, the National Science Foundation, Google, NVIDIA, Amazon, and ARL DCIST CRA W911NF-17-2-0181.

## Footnotes

[2] See Subsection A.1 comparison.

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
