[Supplementary Material · Planning_with_Goal_Conditioned_Policies___NeurIPS_2019_supp.pdf]

# A  Additional Experiments

## A.1  Norm Ablation

We compare using the $\ell_\infty$-norm to minimize the feasibility vector with using the $\ell_1$-norm. As shown in Figure 7, $\ell_\infty$-norm performs better, which matches the intuition it will more consistently push all terms in the feasibility vector towards zero.

Figure 7: We compare using the $\ell_\infty$-norm to the $\ell_1$-norm. We see that the $\ell_\infty$-norm outperforms the $\ell_1$-norm.

## A.2  Optimizer Ablation

We compare the performance of different optimizers on the 2D Navigation tasks. As shown in Figure 8, CEM consistently outperforms other optimizers both in terms of the optimizer loss, and the corresponding final performance on the task.

Figure 8: We compare CEM to different optimizers L-BFGS, Adam, RMSProp, and gradient descent (SGD) that have had their learning rates tuned. (Left) The optimizer loss, where CEM outperforms the other methods. (Right) The performance of the policy after using the plan chosen by each optimizer. We see that the lower optimizer loss of CEM corresponds to a better performance.

## A.3  Likelihood Penalty Ablation

We examine the effect of the additional log-likelihood term (under the VAE prior) in Equation 3. In particular, we vary the weighting hyperparameter $\lambda$ for the 2D Navigation and Push and Reach environments. For each environment, we note the final performance of the RL algorithm, in addition to the log-likelihood values and V values that compose equation 3. See Figure 9 for detailed results. We see that there is a trade-off between achieving a high likelihood under the prior and high V values. As we increase the weighting term $\lambda$ the likelihood values increase while the V values decrease. There is an optimal threshold at which RL performance is maximized. For 2D Navigation, we note this value to be $\lambda = 0.01$ and for Push and Reach any range of values between $0.0001$ and $0.01$. For Ant Navigation, we independently verified an optimal choice of $\lambda = 0.1$.

Figure 9: Examining the effect of the weight $\lambda$ in Equation 3. We note the final RL performance (left), log-likelihood under the VAE prior (middle), and V values (right). As we increase $\lambda$, the log-likelihood values increase while the V values decrease. For 2D navigation (top), we note the optimal value to be $\lambda = 0.01$ and for Push and Reach (bottom) any range of values between $0.0001$ and $0.01$.

# B  Environment Details

## B.1  2D Navigation

The agent must learn to navigate around a square room with a U-shaped wall in the center. See Figure 3 for a visualization of the environment. The dimensions of the space are $8 \times 8$ units, the walls are $1$ unit thick, and the agent is a circle with radius $0.5$ units. The observation is a $48 \times 48$ RGB image and the agent specifies a 2D velocity as the action. At each timestep, the agent can attempt to move up to $0.15$ units in either dimension. The distance for Equation 1 is the distance between the current 2D position and the target position. We note that a greedy policy can easily lower the final distance by moving directly towards the goal. To measure whether or not the final policy performs more non-greedy behavior, we define success as whether or not the policy ends below the horizontal wall and within a diameter of the intended goal. Complete results are provided in Figure 10. Plots are averaged across 5 seeds, with the exception of PETS, which uses 3 seeds due to computational constraints. For image based baselines (all except PETS), we first train VAEs and select the top 5 seeds based on VAE loss. We proceed to training our RL algorithm with one seed per selected VAE. Note that for the ablation study in Figure 6, we select the top VAE seed based on VAE loss, and train our RL algorithm with 5 seeds.

Figure 10: Complete 2D Navigation Results

## B.2  Push and Reach

This task is based on the environment released by Nair et al. [40]. An additional invisible wall around the goal space of the puck has been added to prevent the puck from moving to unreachable hand

locations. In contrast to prior work evaluated on goal-conditioned pushing tasks [2, 47, 8], this task is solved using images as the observations and cannot be solved with a simple, unidirectional pushing behavior [40, 49]. Specifically, the observation is an $84 \times 84$ RGB image showing a top-down view of the scene. The robot is operated via 2D position control, where each action is limited to moving the robot end effector 2 cm in either dimension. The distance for Equation 1 is the Euclidean distance between (1) the goal and (2) the XY-position of the puck concatenated with the XY-position of the hand. We modify the task so as to require the agent to perform temporally extended planning. First, we increase the workspace of the environment to $40$ cm $\times 20$ cm. Second, we evaluate the final policy on 5 hard scenarios which require temporally extended behavior: rather than simply executing a simple, unidirectional pushing behavior, the robot must reach across the table to a corner where the puck is located, move its arm around the puck, and then pull the puck to a different corner of the table, as shown in Figure 3. A trajectory is successful if the final puck position is within $6$ cm of the target position. For context, the puck has a radius of $4$ cm. Complete results are provided in Figure 11. Plots are averaged across 8 seeds, with the exception of PETS, which uses 5 seeds due to computational constraints. For image based baselines (all except PETS), we first train VAEs and select the top 8 seeds based on VAE loss. We proceed to training our RL algorithm with one seed per selected VAE.

Figure 11: Complete Push and Reach Results

## B.3 Ant Navigation

The ant must learn to navigate around a narrow rectangular room with a long wall in the center. See Figure 5 for a visualization of the environment. The dimensions of the space are $7.5 \times 18$ units, the wall is $1.5$ units thick, and the ant has a radius of roughly $0.75$ units. The state includes the position, orientation (in Euler angles rather than quaternions), joint angles, and velocities of the aforementioned components. The gear ratio for the ant is reduced to 10 units, to prevent the ant from flipping over. The distance for Equation 1 is the distance between the current 2D position and the target position, in addition to the differences in orientation of the ant with respect to the target orientation. We define success as whether or not the ant is within $1.5$ units of the goal position. Complete results are provided in Figure 12. Plots are averaged across 15 seeds, with the exception of HIRO, which uses 5 seeds due to computational constraints. For LEAP, we first train VAEs and select the top 5 seeds based on VAE loss. We proceed to training our RL algorithm with three seed per selected VAE. Unlike the image-based experiments, the VAE is not used for training the RL algorithm. It is only used during test time for planning subgoals. The VAE is trained on a dataset in which the ant is in various valid positions of the maze, with a fixed orientation and fixed joint angles.

Figure 12: Complete Ant Navigation Results

| Hyper-parameter | Value |
|---|---|
| Q network hidden sizes | $400, 300$ |
| Policy network hidden sizes | $400, 300$ |
| Q network and policy activation | ReLU |
| Q network output activation | None |
| Policy network output activation | tanh |
| Exploration noise | $\epsilon$-greedy, $\epsilon = .1$ (2D Navigation)<br>OU-process $\theta = .3$, $\sigma = .3$ (Push and Reach and Ant Navigation) |
| # training batches per time step | 1 |
| Batch size | 128 (2D Navigation)<br>2048 (Push and Reach and Ant Navigation) |
| Optimizer | Adam |
| Learning rate (all networks) | 0.001 |
| Target update rate $\tau$ | 0.005 |
| Replay buffer size | 1000000 |

Table 1: TD3 [19] hyperparameters.

## C  Implementation Details

This section contains descriptions and hyperparameters of the experiment implementations.

### C.1  Goal-conditioned reinforcement learning

Both the $Q$ network and policy concatenate all inputs and pass them through a feed-forward network. For RIG, the Q network outputs a scalar corresponding to the infinite discounted sum of rewards. For TDMs, the Q network outputs a vector corresponding to the negative distance between the final state and goal along each of the state dimensions. We train our networks using the twin delayed deep deterministic policy gradient algorithm [19] (TD3). Hyperparameter details are provided in Table 1. When sampling minibatches from the replay buffer, we sample transitions, goals, and times (for TDMs only). For TDM, RIG, and HER+, we relabel the goals in our minibatches in the following manner:

- 20%: original goals from collected trajectories
- 40%: randomly sampled states from the replay buffer
- 40%: future states along the same collected trajectory, as dictated by hindsight experience replay [2] (HER).

We note that in the Ant Navigation task, we split sampling from the replay buffer to 20% from the replay buffer and 20% oracle goals from the environment.

For HER, we relabel the goals in our minibatches in the following manner:

- 20%: original goals from collected trajectories
- 80%: future states along the same collected trajectory

### C.2  Latent space optimization

In this subsection, we describe how we use the cross entropy method (CEM) [11] to optimize equation 3. Given an optimization problem over $K$ subgoals, with each subgoal represented as an $r$-dimensional latent vector, the CEM optimizer is initialized with a standard multivariate Gaussian distribution $\mathcal{N}(0_{rK}, I_{rK})$, where $0_{rK}$ is a $rK$-dimensional vector of zeros, and $I_{rK}$ is the $rK \times rK$ identity matrix. We sample different subgoal sequences from our distribution and evaluate the value of each sample using Equation 3. We then fit a diagonal multivariate Gaussian distribution to the top 5% of samples. We repeat this process for 15 iterations, and at each iteration we sample 1000 subgoal sequences from the fitted Gaussian. For the Ant Navigation task which involves optimizing over significantly higher number of subgoals, we sample 10000 subgoal sequences and run for 50 iterations instead. In addition, we found it beneficial to filter the top 25% of samples for the first half

of iterations, and then filter the top $1\%$ in the latter half. For the weight on the log-likelihood of the latents, we use $\lambda = 0.1$ for 2D Navigation and Ant Navigation tasks, and $\lambda = 0.001$ for Push and Reach.

### C.3 Variational auto-encoder

We use separate VAE architectures for 2D Navigation ($48 \times 48$ image) and Push and Reach ($84 \times 84$ image). For 2D Navigation, encoder kernel sizes of $[5, 3, 3]$, encoder strides of $[3, 2, 2]$, $[16, 32, 64]$ encoder channels, decoder kernel sizes of $[3, 3, 6]$, decoder strides of $[2, 2, 3]$, and $[64, 32, 16]$ decoder channels are used. For Push and Reach, we use encoder kernel sizes of $[5, 5, 5]$, encoder strides of $[3, 3, 3]$, $[16, 16, 32]$ encoder channels, decoder kernel sizes of $[5, 6, 6]$, decoder strides of $[3, 3, 3]$, and $[32, 32, 16]$ decoder channels. Both architectures have a representation size of $16$ and ReLU activation. We trained the 2D Navigation VAEs with binary cross-entropy loss, and the Push and Reach VAEs with mean squared error loss.

For Ant Navigation, our VAE is a generative model for the full state of the ant, rather than images. Our encoder and decoder are multilayer perceptrons with hidden sizes of $[64, 128, 64]$ and ReLU activation. We used a representation size of $8$, and trained the VAE with mean squared error loss.