[Reviews · NeurIPS 2019]

Reviewer 1



Post rebuttal: My suggestions/comments were not addressed in the rebuttal, so I keep my score as is. --------------------------- The authors propose a model-free planning framework using goal-conditioned policies and a value function over a learned compact latent variable representation. Others have proposed this type of two step optimization where one first learns a compact representation with a VAE on randomly collected samples, then use various RL or planning methods on the representation. However, this doesn't work well for high dimensional spaces where random collection of data for learning the representation space does not give enough samples -- especially from the optimal policy. This work doesn't address this issue, by only evaluating on environments with very small state spaces, where random sampling to train the VAE is feasible. Originality: The idea of planning using TDMs over a latent representation is novel, and a promising direction for goal-directed planning in high-dimensional observation spaces. Related work in neighboring areas is adequately cited. Quality: The authors provide a theoretically sound justification for their method, and evaluate in both 2D navigation and a robotic manipulation simulation with higher dimensional latent space and show ability to perform better at long term reasoning compared to existing model-based and model-free methods. Clarity: The paper is clearly written, with contributions and novelty clearly stated, and experiments, hypotheses being tested, and ablations clearly described and justified. Significance: The results are somewhat important, in that they clearly show higher performance with reasoning in latent space using TDM compared to planning with MPC, model-free methods, and one method with imagined goals. However, it would be also useful to see how different training methods compare when using this method, as training on randomly collected samples from the environment will fail on complex tasks. Even an example of a more complex task in which this method fails because of lack of exploration, or trying an iterative training procedure would be useful to see and discussion of this failure mode.

Reviewer 2



In order to overcome the limitation of RL methods in learning temporally extended tasks, this paper proposes to combine a planner, which works with a high-level abstraction of the problem, with a model-free RL approach that learns goal-conditioned policies, which provide such an abstraction to the planner. In particular, the authors focus on goal reaching tasks with high-dimensional observation and goal spaces, like images. The proposed approach is then compared to simple model-free (TDM) and model-based (MPC) approaches and a state-of-the-art method (RIG) on two imaged based tasks. Although the idea of combining model-free and model-based approaches is definitely not new, the proposed approach, as far as I know, is original. The authors have done a good job in placing their contribution with respect to the state of the art. The technical content of the paper appears to be correct. The paper is well written and clearly organized. My main concern with this paper is about the significance of the contribution. The paper lacks any theoretical analysis of the proposed approach (no sample complexity, no computational complexity, no convergence guarantees). So the contribution can be evaluated only from the empirical analysis. I do not think that the two considered domains, even if not trivial, are enough to properly evaluate the effectiveness of the proposed approach. The advantages with respect to the considered (not so strong) baselines are not very large, especially considered the additional complexity introduced by the proposed approach. The number of runs used in the experiments is quite low and I have found no explanation of the meaning of the shaded areas (even if the authors claimed to have specified it in the reproducibility checklist). Are they confidence intervals or standard deviations or what else? Furthermore, it would be interesting to see a comparison with respect to other model-free and model-based deep RL approaches (like DQN and Guided Policy Search). I do not think that the contribution introduced in this paper is so strong to attract the interest of many researchers in the field. Minor issues: l.237 The initial arm and and pack locations => The initial arm and pack locations ========================== Post Rebuttal The authors have done a good job of addressing the issues raised in the reviews. The additional results show that the proposed approach outperforms other baselines. Although I still feel that the significance of the contribution is quite low, I have decided to raise my score to 6.

Reviewer 3



The paper proposes to combine model-free RL for short-horizon goal reaching with model-based planner over a latent variable representation of subgoals. The method is based on TDMs which decompose trajectories into several subgoals that are connected by the lower-policy. The higher-policy can utilize the lower policy as the model to plan a trajectory to the goal by optimizing intermediate subgoals. To overcome the high-dimension problem in the image domain, the paper reduces the states' dimensionality with VAE and solves the optimization problem on the latent domain. The paper is clear and the experiments are sound. However, the idea of planning with model-free policy has already been discussed by TDM; using VAE to plan over images is also discussed in the literature. The paper simply combined these two, which is incremental to me. The paper missed several recent references, e.g.: Learning Plannable Representations with Causal InfoGAN Learning Actionable Representations with Goal-Conditioned Policies =========================================================== The authors addressed most of my concerns, so I increase the score to 6. Although the significance of the contribution is still limited, it will be a good step towards understanding the connection between modeling and planning in RL settings.

[Author Response · NeurIPS 2019]

We thank the reviewers for their helpful comments. To address **R2** and **R3** concerns, we modified the manipulation
task to be more challenging and better test generalization. We added two new state-of-the-art baselines (PETS [1] and
HER [2]). We also present preliminary HIRO [6] comparisons, as requested by **R3** . To address **R3** , we added an
experiment studying different optimizer choices. These additional experiments should address the primary concerns
raised by the reviewers. We summarize the important points below.

Figure 1: We made Push and Reach more challenging and added new baselines (PETS, HER). MPC results are omitted for clarity.

**R2** , **R3** : Regarding additional comparisons, the baselines now include PETS, a model-based method, and HER, a
goal-conditioned method. The model-free TDM in the submission is already trained on the VAE state representation, as
**R3** requested. Figure 1 shows that LERP significantly outperforms these methods.

**R3** : To address questions of generalization, we have modified the Push and Reach task. We now varied the initial state
configuration on the pushing task during test time to include 5 rather than 1 challenging configuration. The new Push
and Reach experiment shows that LERP significantly outperforms prior methods at generalizing. Specifically, Figure 2
shows that only LERP can solve all initial configurations, while the next-best method (TDM-100) fails to consistently
solve any of them. Due to time constraints, we did not have time to run the "different block" configuration requested,
and exploring generalization to new environment configurations would be an interesting avenue for future work.

Figure 2: (Left) If we split the next-best method (TDM-100) by test configuration, it fails to generalize to all configurations. By contrast, we see that LERP solves all configurations. (Right) We compare different optimizers for LERP. We found that CEM outperformed L-BFGS, Adam, RMSProp, and gradient descent (SGD) even after tuning the learning rate.

**R3** : We compared to gradient-based optimizers after tuning their learning rates.
Figure 2 shows that CEM consistently performed the best, likely due to its ability
to escape local optima. Using more advanced non-gradient optimizers would be
promising future work.

**R2** : We increased the number of seeds from 3 to 8, for Push and Reach. The
shaded region represents one standard deviation across seeds. We will update
figures accordingly and describe the shaded region in Section 5.

**R3** : We also compared to HIRO on the 2D Navigation. Due to time constraints,
we were only able to run one seed, but the preliminary results in Figure 3 suggest
that the existing baselines and our method signficantly outperform HIRO.

**R3** : Thank you for the additional references. We will add a discussion to the
Related Works section.

Figure 3: Preliminary HIRO results HIRO on 2D Navigation. Compared to Figure 1 (left), HIRO does not appear competitive.

[1] Chua et al. Deep Reinforcement Learning in a Handful of Trials using Probabilistic Dynamics Models. NeurIPS 2018. [2] Andrychowicz et al. Hindsight Experience Replay. NeurIPS. 2017. [3] Florensa et al. Automatic Goal Generation for Reinforcement Learning Agents. ICML 2018. [4] Pong et al. Skew-Fit: State-Covering Self-Supervised Reinforcement Learning. CoRR 2018. [5] Zhao et al. Energy-Based Hindsight Experience Prioritization. CoRL. 2018. [6] Nachum et al. Data-Efficient Hierarchical Reinforcement Learning. NeurIPS. 2018.

[Meta-Review · NeurIPS 2019]

There is general consensus that the paper is well executed but its significance may be low. Yet, the reviewers believe that the elements provided in the rebuttal can make the contribution more solid. I encourage the authors to integrate the rebuttal in the final version and follow the recommendations of the authors.